# Gardening in Ashes: The Possibilities and Limitations of Gardening to Support Indigenous Health and Well-Being in the Context of Wildfires and Colonialism

**DOI:** 10.3390/ijerph17093273

**Published:** 2020-05-08

**Authors:** Kelsey Timler, Dancing Water Sandy

**Affiliations:** Interdisciplinary Studies, University of British Columbia, Vancouver, BC V6B2T5, Canada; dancing.water.lulua@gmail.com

**Keywords:** Indigenous health, reciprocity, gardening, climate change, climate crisis, colonialism, Indigenous sovereignty, wildfires, natural gardens, food security, blanket exercise

## Abstract

In this paper, we will discuss gardening as a relationship with nature and an ongoing process to support Indigenous health and well-being in the context of the climate crisis and increasingly widespread forest fires. We will explore the concept of gardening as both a Euro-Western agriculture practice and as a longstanding Indigenous practice—wherein naturally occurring gardens are tended in relationship and related to a wider engagement with the natural world — and the influences of colonialism and climate change on both. Drawing on our experiences as an Indigenous Knowledge Keeper (Dancing Water) and a non-Indigenous community-based researcher (Kelsey), our dialogue will outline ways to support health and well-being through land-based activities that connect with Indigenous traditions in ways that draw on relationships to confront colonialism and the influences of climate change. This dialogue is founded on our experiences in the central interior of British Columbia, Canada, one of the areas hit hardest by the 2017 wildfires. We will explore the possibilities and limitations of gardening and the wider concept of reciprocity and relationship as a means to support food security, food sovereignty, and health for Indigenous Peoples.

## 1. Introduction

Indigenous Peoples in Canada have diverse and complex relationships with their ancestral lands that have sustained them across a wide range of ecosystems since time immemorial [1,2,3,4]. These complex understandings of place are founded on the knowledge of reciprocal benefit with, and responsibility to the natural world, and are practiced through complex foodways that support social cohesion and relationships between individuals, families, communities and the natural world [2,5,6,7,8]. Historic and ongoing colonialism has resulted in the purposeful disruption of many of these land-based practices [9,10], including forced relocation through the reserve system, the regulation of subsistence practices, and policy-enforced economic vulnerability and related barriers to subsistence, such as the cost of gas, transportation, and tools needed for hunting and fishing [11]. This “culinary imperialism” was furthered forced dietary changes in residential schools and Indian hospitals [12], and ongoing colonial encroachment through industry, agriculture, and urban development [3]. This colonial context has created and sustained deeply entrenched health and social inequities that impact many Indigenous communities in British Columbia (BC) and across Canada [5,13,14,15,16,17,18]. These barriers, which include food insecurity and socio-economic vulnerability [19,20,21,22], are worsened by the climate crisis [23], as many communities rely on the seasonal availability of animals, fish, and plant foods and medicines for their health and wellbeing, both in terms of physical sustenance, belonging, and ceremony [24,25]. In this context, a great deal of Euro-Western academic research has aimed to address food insecurity for Indigenous Peoples through the development of gardening programs. However, there remains limited attention to the challenges and opportunities facing Indigenous gardens and other ancestral foodways in the context of colonialism and climate change [25,26]. The possibilities and limitations of gardening to support health and wellbeing are highlighted by a harsh and unforgiving season of wildfires across BC in 2017 [27]. These fires burnt over 1.3 million hectares of land, displacing 65,000 people, and costing $568 million dollars in suppression efforts. The Cariboo-Chilcotin region in the interior of the province was impacted by two of the largest fires, which encompassed 63% of the total impacted land in BC [27]. This region is the unceded and ancestral lands of the Tsilhqot’in, Dakelh, and Secwépemc peoples, which include several rural and remote communities facing ongoing colonial barriers to health and wellbeing. Against this background, we will explore the possibilities and limitations of gardening as a means to support food security, food sovereignty and health for Indigenous Peoples. Drawing from our experiences as an Indigenous Knowledge Keeper (Dancing Water) and a non-Indigenous community-based researcher (Kelsey), our dialogue will outline ways to support health and well-being through land-based activities that connect with Indigenous traditions in ways that confront colonialism and the influences of climate change. Through this dialogue we will explore and expand upon the concept of gardening as both a Euro-Western agricultural practice and as a longstanding Indigenous tradition—wherein naturally occurring gardens were tended in relationship—and the influences of colonialism and climate change on both. The foundational importance of reciprocity and relationship in the context of foodways to support holistic health and wellbeing will be outlined.

## 2. Background

The benefits of gardening are well known within Euro-Western scientific literature, including the benefits of simply being outside in nature [28], as well as lowered cortisol levels, impacts on social cohesion, sense of pride, and increased food security [29,30,31]. The benefits of gardening—as an agriculture practice and hobby—are also evident when focusing specifically on communities who face social, cultural and/or economic marginalization [32,33]. For instance, findings from a study on the impacts of a community garden in a socio-economically marginalized neighborhood in Toronto found that participants experienced stress relief, increased food security, and had opportunities to build relationships across cultures through the preparing and sharing of food [33]. While diet and nutrition research often focuses on food security and related interventions, such as gardening, the wider interrelationships between land, food, the natural world, health, and wellbeing have been recognized by Indigenous Peoples since time immemorial, with relationships and reciprocal benefit with the natural world often foundational across diverse worldviews [1,2,5,6,34,35]. This knowledge is embodied, lived and valid, based on ancestral and ongoing understandings of weather patterns, seasonality, and the relationships between a wide range of plants and animals. While the reliance of some Indigenous Nations on salmon populations and wild game, such as deer and moose, are widely recognized in wider Canadian society, many Indigenous communities have historic, ongoing and evolving gardening and food gathering practices [3]. These include clam and estuarial root gardens and camas and berry gardens, as well as a wide variety of other land-based and aquatic plant foods [3,36,37,38]. In some instances, Euro-Western science has only just recognized these longstanding relationships to food sources, despite ongoing evidence from within Indigenous knowledge systems [39].

Diverse Indigenous foodways developed alongside ecosystem-specific knowledge, and continue to be transferred across generations through oral tradition, trade, and ceremony [5,7,8]. Colonial settlers failed to acknowledge Indigenous land rights and relationships [3], and while new food items were pragmatically adopted into Indigenous diets, colonial land theft and regulation eroded food sovereignty [9,10], as did industry, pollution, over-extraction, and invasive plants [40]. While Indigenous Peoples have and continue to tend plant foods and medicines in evolving, complex, and relational ways, these ceremonial and medicinal practices have been negatively impacted by colonialism, capitalism, and resource extraction [4,9,10,40,41,42,43,44], all of which are further tied to climate change [25,45,46,47,48,49,50]. These systems and structures are embedded in Western ideals of economic production, neoliberal individualism, and the far-reaching and often un-problematized benefits of large-scale resource extraction and related technological innovations.

Climate change adaptation and mitigation have been highlighted as essential for Indigenous communities, whose ongoing experiences of colonialism increase their experience of harms related to the climate crisis and correlated industry, harms more explicitly felt by Indigenous women, children and gender non-conforming peoples [27,51]. Yet, in the face of these persistent colonial barriers, communities continue to protect their land and waters, food sources and health and wellbeing, as evidenced by the ongoing work of land defenders and protesters across the nation [47,48,49,50]. Dawn Morrison, founder of the BC Food Systems Networking Group on Indigenous Food Sovereignty, notes that “the best way we can defend our grandchildren’s future is to protect, conserve and restore the health of the forests, fields and waterways where we hunt, fish, farm and gather our food” [52]. This defense of future generations extends beyond food security, including confronting the complex barriers to not only physical health and diet-related disease [19,20], but emotional, spiritual and mental wellness [21,22]. Barriers to the social and historic determinants of health, including food security and social cohesion, have been correlated with disproportionate rates of Indigenous youth mental illness and suicide, one of many examples highlighting the urgency of supporting Indigenous foodways as more than mere nutrition [53,54].

Gardening has been proposed as one way to address food insecurity and support health for Indigenous Peoples in BC and across Canada [55,56], and several communities have undertaken gardening projects to support dietary health [55,56]. While gardening can provide increased access to fresh vegetables, the potentials of individual and community gardens need to be taken into account within the context of historic and ongoing colonialism, and the correlated barriers to health and wellbeing facing many communities. Additionally, the concept of gardening within academic literature is largely Eurocentric, rarely accounting for ancestral and ongoing Indigenous gardening and food practices and the inextricable linkages between food, identity and belonging that have supported Indigenous Peoples for millennia [57]. In this context, Euro-Western understandings of gardening have flattened and simplified the complex relationality fundamental to many Indigenous foodways, resulting in an often-narrow focus on growing fruits and vegetables as a means to support food security, without addressing the wider holism called for across Indigenous health and food systems [1,2,4,10,57]. As the impacts of colonialism and extractive capitalism continue, many communities are working to revitalize language, foodways, and social structures that support intergenerational knowledge sharing and cultural continuity [58,59], with keen attention paid to the empowerment of children and youth [60], and to the interconnections between individual, community and environmental health [23]. Gardening is one such example [56,61,62].

In 2017, wildfires ravaged the central interior of BC, resulting in what was at that time one the worst wildfire seasons in Canada’s recorded history [27]. In this context, our dialogue will trace the history of gardening from both Indigenous and settler contexts, exploring opportunities for collaboration and points of tension for gardening in the context of climate crisis and colonialism. Through our dialogue, presented as a conversation to align with the relational aspects of Indigenous food systems discussed in this paper, we will explore how widening our understanding of gardening, in the face of climate change and colonialism, can support health and healing for Indigenous and non-Indigenous peoples in Canada.

## 3. Dialogue as Method

We are committed to Indigenous, decolonial and relationship-centred ways of being across research and practice. In both of our work, accounting for context, history, and the diversity of Indigenous Peoples, places, and ways of knowing necessitates that “there is no one set of steps or practices” that can dictate knowledge development [63], and dialogue as method allows for responsive and reciprocal engagement between story-tellers and story-learners [64,65,66].

Dialogue and storytelling have become more and more widespread across disciplines as the importance of inclusivity and engagement in human subject research is being increasingly recognized by institutional research ethics boards and funding bodies [67,68,69,70,71]. Dialogue and storytelling as method are widespread across qualitative studies, and is often foundational to community-based research conducted in partnership with Indigenous Peoples [72,73]. This method aims to confront power imbalances within academia, pushing science to exist beyond a “privileged discourse” [74,75] which often relies on “escoteric writing style[s]” [74]. In many disciplines, inaccessible language exists as a “a mechanism of exclusion” [76] which in turn distance research findings from applied community benefit. Without careful attention, alternate knowledge holders and systems are systematically marginalized [75,77,78,79], perpetuating inequities and cognitive injustice [74,77]. Dialogue as method and Knowledge Mobilization practice offers the potential to uphold the voices pushed to the margins, and unmask and deconstruct power [80] all while honouring Indigenous oral traditions and knowledge sharing practices [65,81].

Western scientific knowledge is assumed to be objective, born from reductionist, positivist and linear understandings of time and hierarchies of knowledge. Despite a wealth of Indigenous scholarship [34,72,82,83,84,85,86], Indigenous knowledges are seldom integrated into policy and practice. This has widespread impacts on health, social and environmental inequities in Canada. Today, Indigenous Peoples face inequities reified in discriminatory treatment across resource extraction, education, health, criminal justice, employment, and social service sectors [87,88,89]. Despite tireless work by Indigenous scholars, Indigenous ontologies and epistemologies, and concepts of health, wellbeing and environmental management [82,86,90,91,92,93,94,95,96,97] are largely absent from social consciousness, research and practice, which in turn impacts the environmental health and wellness of Indigenous communities.

Dialogue as method aligns with the Canadian Tri-Council Policy Statement on Research with Indigenous Peoples [72], the Indigenous OCAP principles (Ownership, Control, Access and Possession) [98], and the four Rs from First Nations Education within research: Respect, Relevance, Reciprocity, and Responsibility [99]; frameworks which provide outlines upon which relationships can be nurtured [72,73]. Dialogue as method provides opportunities for relationships to be reflected throughout the Knowledge Mobilization process. Privileging the voices of Indigenous Knowledge Keepers [63], centering diverse cultural practices and ceremonies [63,100,101], and creating spaces to share food and support reciprocity [101] are central to research conducted with and for Indigenous Peoples. In this context, knowledge sharing focused on addressing climate change and colonialism requires a keen awareness of history and context, and the use of dialogue to support reciprocity and relationship.

While we bring our teachers and mentors with us into this dialogue, our conversation is intended to share our perspectives as individuals. We are individuals engaged in overlapping communities, who have learnt and are learning from the diverse Knowledge Keepers, Elders, academic professors, family members and friends that surround us. Dancing Water does not speak for all Indigenous peoples, all Secwépemc and Cree people, or even all members of her community of Sugarcane, BC. In the same way, Kelsey does not speak for all Euro-Canadian setters, all doctoral students, or all community-based researchers. Throughout our dialogue we will layer in other voices, though recognize that the depth of knowledge that has been shared with us is innumerable and complex in ways that expand beyond the scope of this paper.

## 4. Dialogue

*Kelsey*: I’m really excited have the opportunity to talk about gardening and foodways with you Dancing Water! Should we begin by introducing ourselves?

*Dancing Water*: Yes, of course! I am of the Secwépemc and Cree Nations, in addition to having Scottish ancestry. I carry traditional knowledge that has been entrusted to me from my family, passed down from my ancestors, and I am committed to using that knowledge to support my community. I often say that community and family are synonymous to my way of being, and I consider it my responsibility to share these teachings of Indigenous epistemologies and pedagogy through the lens of land-based learning and healing. My commitment to decolonization and healing for Indigenous Peoples in Canada across the intersections of health, justice, and education are also foundational to my work as a First Nations Curriculum Development Teacher in the central interior of BC. Also, I love spending time on the land harvesting plant foods and medicines, fishing and hunting with my son.

*Kelsey*: Thanks Dancing Water! As for me, I’m a settler with Swedish and Polish ancestry. I’m originally from Treaty 7 territory in Calgary, Alberta, but have been living in Vancouver BC, on the unceded territories of the Squamish, Musqueam and Tsleil-Waututh Nations for over a decade, and have fallen in love with this beautiful part of the world. I have always enjoyed preparing and sharing food, working as a professional cook for many years before getting interested in the socio-economic and historic aspects of the food system. Today my work is focused on supporting food justice for people in BC, and I have been working with Indigenous communities around food sovereignty. And it’s through this work that we met! I was spending time in the Tsilhqot’in First Nation working on a community-based research project focused on healing and reciprocity for Indigenous Peoples who are incarcerated. In a nutshell, this project, led by Dr. Helen Brown in the University of British, School of Nursing, focuses on learning about a federal corrections initiative where incarcerated people build and gift things to the Nation, including the growing and gifting of organic produce (and more information can be found elsewhere [57,95]). Through our ongoing relationship building, a foundational part of community-based research, we ended up walking into this beautiful, colorful classroom in one of the Tsilhqot’in communities, to spend some time with kids, and there you were.

*Dancing Water*: Yes, I met you and Helen during Culture Week 2018. Culture Week is an annual event in BC’s School District 27 where the Xeni Gwet’in Nation supports children from that community and the surrounding area to engage in land-based learning, things like harvesting mountain potatoes and learning horsemanship. I was teaching there at that time.

*Kelsey*: I remember asking you if we could store our art supplies in your classroom, we were doing art workshops to support ongoing relationship building and learn more about the connections between art, identity, healing and reciprocity, in the context of this wider project where incarcerated men were making beautiful arts and crafts and growing vegetables and gifting them to the community. Anyways, that’s a bit of a tangent.

*Dancing Water*: Maybe, but also important to start at a place of relationship. That’s how we know each other. And since then we’ve connected in different areas—I’ve come to UBC to teach Helen’s nursing students how to make Indigenous teas, and the three of us have travelled to Ottawa together to learn about Indigenous-led land-based healing programs taking place across the country. It’s about those connections. And when we think about Indigenous gardening, berry patches, crab apple trees, different barks, they are communally accessed and utilized, they are relationships.

I was taught as a young person to show the plant some love so it produces next year. And this occurs out there on the land, you can tell where people have picked, based on where things have been groomed and taken care of, by communities.

And these community formations and territories are historic and longstanding, created alongside land formations, rivers, mountains, and relationships with neighbouring peoples. For example, Secwépemc, meaning *the spread-out people,* had pit houses, and they leave these indentations on the land, evidence that they were there. They sometimes appear in other territories, and it makes people really uncomfortable, because colonial judicial processes have made it so that any Nation with a territorial claim to their land cannot have disputes with other Nations. Today, if First Nations are going through overlap disputes, they can’t enter into treaty processes, or court proceedings including matters of rights and title. But historically there was overlap, because of trade, disease and wintering with other people, relationships that helped us survive. Another example is at Sheep Creek, a place not far from where I live, there are pit house indentations, though many Tsilhqot’in believe that’s not a pit house, Elders say ‘they did come over, their tribe was killed off by small pox, and they came over and we accepted them.’ You can’t deny that Secwépemc people were there. People moved before colonialism, and then disease forced more movement after colonial contact. The movement of people and the sharing of space is inevitable, which means that creating relationships is inevitable.

*Kelsey*: I love the way you’ve framed the creation of relationships as inevitable across these histories, even amongst the real tragedy and violence that colonialism has and continues to perpetuate. Colonialism disrupted Indigenous Peoples’ relationships with specific areas of land not only because it created barriers to being on the land, like the reserve system and the creation of ‘private property,’ which is what I usually read about, but also because the way that communities were thought of and bounded was changed, is that right? The forceful confining of communities on reserves was mandated by the Indian Act, and that would impact access to different plant ecosystems, as well as relationships between neighboring Nations.

*Dancing Water*: Yes, and that is highlighted by the history of my own community, the T’exelc, which means *charging up*, if you think about water rushing up, also known as the Williams Lake Indian Band. Colonial settlers stole our territories and in return gave us a few hundred acres of rock, land so unproductive it was unsuitable for settlers. There’s a famous speech by Chief William in 1879, where he said “the white men have taken all the land and all the fish” and that “we have nothing to eat. My people are sick. My young men are angry.” He goes on to talk about how colonial agriculture had disrupted the careful relationships between humans and the natural world. He talks about being given a bag of turnip seed, but having no land to plant the seeds. And the impact of this has been incredibly significant. Have you ever done a blanket exercise?

*Kelsey*: No, what’s that?

*Dancing Water*: They were developed by Kairos Canada, an ecological and human rights coalition. It was developed in response to calls for education on Indigenous histories in Canada outlined in the 1996 Report of the Royal Commission on Aboriginal Peoples [102]. The Exercise covers over 500 years in a 90-minute experiential workshops that “aims to foster understanding about our shared history as Indigenous and non-Indigenous peoples” [103], and since then it’s been updated to align with the Truth and Reconciliation Commission of Canada’s final report [104]. You layout blankets on the floor to represent different Indigenous territories, and you stand on them and go through history as blankets are taken away, one by one, you and the rest of the group are standing on smaller and smaller sections of blankets, until eventually you’re all crowded on one, and there isn’t enough room. People are also removed from the blankets, to represent those that died from disease, starvation, the Indian Residential School System and other forms of colonial violence. There is an exercise developed for both national and provincial histories, and they are very powerful visual experiences, to start to understand the impact of colonialism on our relationships and access to the land.

*Kelsey*: That sounds incredibly powerful.

*Dancing Water*: It is. But it’s also not the whole story. Indigenous Peoples are strong, and our relationships continue. For instance, I have friends who live in Vancouver, and they come and visit me up here to go hunting, and when they do, they hunt for themselves, for their families and for ceremonies, but also for my son and I, which in turn we share with our family. So, we trade. They will hunt up here, but they may also bring salmon or crab, things that we don’t have access to. Another example is my late Great Auntie Laura from Soda Creek, who has a crab apple tree in her yard. Before I go there to pick, my Mom and I ask permission from her son, to show respect, and we ask if he needs any apples picked. Sometimes we bring a gift, a jar of jam or something else homemade.

*Kelsey*: So those complex, relationship-based trading systems are still in place, to some degree. It sounds like you are continuing those practices through your individual and kinship networks, is that happening at the Nation to Nation level?

*Dancing Water*: It’s mostly on an individual basis, though sometimes trading is done Nation to Nation. For example, Sto:lo and Musqueam will give us salmon when we can’t get them, because of a rockslide into the Fraser river, or impacts of mining, or wildfires—things that impede the fish or our ability to go to the river. This year my Mom got three sockeye and three pink salmon because she’s an Elder. But salmon stocks are diminishing everywhere, so they can never send enough to fully meet community needs. So, growing our own food, whether Indigenous or not, can help to meet some of those needs. There are ways of making cultural practices part of modern gardening, without impeding on our beliefs or the plants themselves. And sometimes it’s not only about the resource. Some years ago, I was feeling frustrated and tired because of things happening in my life, and my cousin saw that and invited me on an outing to catch trout with her family. So, we went out, and the kids ran around and we had a picnic and fished in the creek, and there were eagles who were also harvesting trout from that creek. I felt better, not just in the short-term, but in a healing way, surrounded by family and by nature, I was being reconnected. I brought baked goods to share, and we spent time together. And it wasn’t about the trout, the small fish were not what I was missing, but the relationships with the creek, the eagles, and my family.

*Kelsey*: That’s beautiful, I have an image in my mind of the creek and it’s beautiful.

*Dancing Water:* It is beautiful, and that wellbeing we derive from nature needs to be met with responsibility and care from our end as well. It’s trade, between us and the natural world. So, back to my late Great Aunt’s crab apple trees, when I’m done picking what I needed, I groom the tree and the area—clear and pick up dead and broken branches and other debris. This is the same practice if I’m picking Saskatoon’s or chokecherries when you’re done picking you pull away the weeds and dead branches, anything that could impede growth (see: Figure 1). You want the plant to be there next year, to do its job, to feed you, so you clean up. Much like you would in your (Western) garden, you weed and prune everything so the resources go to the plants that are going to produce. This is a teaching that goes back before colonialism.

*Kelsey*: Euro-Western gardens are controlled, so I can picture what pruning means in that sense, but to prune and manage Indigenous gardens makes sense, to help ecosystems thrive.

*Dancing Water*: Yes, but not everyone has the traditional knowledge to support harvesting practices, and our resources are being used more and more by non-Indigenous people who aren’t aware. They harvest the resources often, without taking the time to care for the plant. So, if you see people who don’t know the practice you tell them, you share the teachings. Sometimes that goes well, and sometimes it doesn’t. They are taking a resource, but to us it’s not merely a resource, it’s *a place*. We look at it as a whole, not just a berry or apple.

*Kelsey*: That’s such a contrast to Euro-Western gardening, which is about the control of the land. At one time gardening in Europe was considered an art-form, the pruning and management of plants for solely aesthetic purposes, and was something done by hired experts. Today it’s still about control, often still the rows of one vegetable followed by the rows of another, and taking all or most of what the plant has to offer is considered the norm.

*Dancing Water*: It’s a different relationship with nature. There is still artistry, but it’s based in reciprocity, not control. For example, prior to harvesting from an area, we do an offering to thank the Creator and the plant, because the Creator gave us the gift of life for another day and the ability to travel out on the land safely. We thank the plant because it is ultimately sacrificing a lot of itself. Harvesting is not simply an act of taking, it is reciprocal and heavily embedded within cultural practices. Control isn’t the objective, it’s about gratitude and an expression of interconnected relationships with the land.

*Kelsey*: So, given all of this, what do you think is important when talking about expanding and revitalizing Indigenous foodways?

*Dancing Water*: I think we need to teach our people that that is our food. The pizza you buy in a box is not your food, it’s not good for you. We need to simultaneously take multiple generations of people out on the land, because not everyone has that knowledge anymore. I’m really fortunate that my Aunties kept pieces of knowledge, and that I moved back home and was able to go out on the land. To revitalize Indigenous foodways we need to be on the land. These foods are used in ceremony, they are used to celebrate people’s lives, to bring people into the world, to celebrate milestones and to care for people. And if we don’t harvest them, they won’t be there for these events. We need community initiatives to create awareness. I often take nieces and nephews out, I take everyone out when I’m home – picking rosehips, Labrador tea. Wherever I go, I bring someone, it’s engrained in me. And whatever they give me, fuel money, some of their harvest, a story, a prayer, that’s payment enough. Those families are getting what they need because you’ve brought them. So that’s another layer of relationship. Coming back to the trout fishing trip I went on with my cousin, she saw I needed help, so into her truck we went. My trade that day was a story, my story of hurt, and at the end of the day, I was full of hope.

*Kelsey*: It’s not just what you’re harvesting, but how you’re doing it and who you’re doing it with.

*Dancing Water*: Yes. And where you’re harvesting. We have found that people from other Nations are picking the resources from our trees, the thinking is ‘oh there are lots at the Sugarcane reserve, and it’s close, so let’s just go there.’ Sugarcane Elders will go to pick their berries, wild cranberries or chokecherries, and they’re all gone. And it’s because people from town, from other Nations, will drive to Sugarcane and harvest it all. So cultural norms to ask for permission aren’t being respected. It’s becoming about the berries as a resource, not as relationship. Our community leaves those resources for our Elders with limited transportation, and it’s insulting to then have them harvested by non-community members. For our Elders to go out onto the land and to see that the plants have been picked over creates a heavy feeling in their hearts. That’s a natural law being broken, a respect that has always been there, but now is changing.

*Kelsey*: So that colonial focus on the individual, instead of the community and the natural world as intertwined. Respect for Elders, for the land, for each other, it’s all connected.

*Dancing Water*: When you return to the land, it’s about how all those teachings intertwine, it’s a full circle. It’s what will heal us. It really inputs all those things you need to be a good community member. It’s about food as connection. In my classroom there’s always food, the youth don’t need to ask for it, and there’s no policing of who takes what or how much they take. There’s an automatic respect for fellow youth and to make sure that what you take reflects the needs of others. Having to ask for food is degrading. Food should be used to build connection and inclusion.

*Kelsey*: You’re expanding beyond food as nutrition and diet-related health to food as an aspect of identity and social inclusion and belonging. And that’s one of the things I love about foodways, that there is such potential for connection. Because we all eat, regardless of our ethnicity or religion or sexuality, it’s something we all share. And that connection is such an opportunity for learning how to be together, what you just said—about students taking in a way that reflects other’s needs—is such a clear example of equity, something that the global colonial food system does not reflect. I wonder if you can talk more about the policing of food, because I think there’s a lot of important history there, and am interested to hear your thoughts.

*Dancing Water*: Sure. Policing food is a highly controversial topic for many Indigenous Peoples, when people are policing food in our community it means there isn’t enough. And it all links back to colonialism and the Indian Act, which broke our natural laws and made it so that these important teachings that sustain us and the natural world, weren’t able to be passed down, or not in the same ways. And so, our own people began to police our food sources, maybe as a way to have a sense of control. Some people in my community want to put fences and padlocks around berry patches and other harvesting areas, to make sure those plant foods and medicines are still there for our Elders and communities. I’m all for protecting our territories, but it makes me uncomfortable, when it’s about policing; that’s not our way.

*Kelsey:* It’s so complicated, because that regulation of food sources was, and still is, one of the main methods of colonial control, from hunting and fishing regulations to confinement through the reserve system and the violent disruption of foodways in residential schools and foster care. And it’s all driven by this framing of food and land as resource and commodity, the complex relationality of plant foods reduced to monocrop agriculture. But it makes sense that some Indigenous folks will take that up, as a way to confront colonialism.

*Dancing Water:* It does, it’s wanting control, and it’s seeing the resources being over-harvested and not knowing what else to do. But it’s not in line with our teachings. For example, when my Dad passed, based on my community’s teachings I was not allowed to hunt, fish, or harvest for a year. My family and community came together to support me, to care for me through food and provide for my family. Our families honour this teaching and look out for those who have deaths in their inner family. That’s not policing food, but relationship and community.

*Kelsey*: That’s such a power example of health and belonging through food. And what happens when you try and put a padlock around a teaching like that? I can see the discomfort, between trying to protect resources in the context of colonialism, where so many people see nature as an inexhaustible resource—despite strong evidence to the contrary—but then wanting to honour natural laws and relationships. That’s really complicated, and I’m wondering what non-Indigenous people like myself can do to support the safe keeping and revitalization of these foodways? 

*Dancing Water*: I think awareness of the history, that colonialism exists, and that it has disrupted our relationships to the land and to each other, and that those interconnections need to be rebuilt, not just for Indigenous Peoples, but for the global community to survive. My community of Sugarcane has a highway and railways running through it, there’s cell towers, gas lines, powerlines. So, if you think of the tiny little reserve that we’re left standing on, if you think of the blanket exercise—you’re all on half a blanket, and they have impeded us even more through intensive infrastructure and industry. And then in our wider territory we have the Mount Polley mine, which broke into our water system and poisoned our water and fish. Every type of industry has happened in our territory. If you think about colonialism, it’s the regulation of our relationships with land to serve industry. We are told where we can hunt, who can hunt, when we can fish, and yet those rules aren’t protecting the animals. Our Peoples have had to police the hunt of cow moose, because people weren’t respecting the law of the land, and shooting cow moose, which impacts future generations. Just look at the Wet’suwet’en in northern BC, fighting to protect their water and their land, and having the police blocking them from accessing their food sources. These food sources aren’t secondary, they are primary resources. If I’m grooming our natural gardens, to support the resource to continue to grow, I’m doing that with the intention of returning to harvest. It’s not just blueberries, it’s an ongoing relationship with that plant, and that ecosystem, and the other animals that use that same area. I think it’s important for non-Indigenous people to include Indigenous Peoples in their harvesting process, and understand and honour our time and knowledge.

*Kelsey*: I completely agree, though would love for you to say a bit more about including Indigenous Peoples in harvesting. There’s been a lot written about the emotional labour and burden that black, Indigenous and People of Colour have been forced to bear when white folks want to learn more about how not to be racist, or ways to support decolonization, or, in this case, how to harvest responsibly. Great places to start include articles by Threads of Solidarity [105], John Metta [106] and Zoe Matties [107]. And I know that’s not what you mean at all! But I want to make clear that what you’re saying is about more than asking any Indigenous person to teach you. There’s a lot of amazing resources out there (including academic publications, websites, toolkits and Indigenous climate action strategies), and as a settler it’s important to educate yourself on the history, and make sure you’re aware of the privilege of accessing these resources. Engaging with Indigenous communities is important, but it can’t be extractive, and it’s our work—the white people of Canada —to come to relationships in humble and informed ways.

*Dancing Water*: Any engagement should be done as part of the development of a relationship, to the land and to one another. There’s a teaching in beadwork that I think is relevant here, and that was told to me by Nelson Leon, a Knowledge Keeper from the Chehalis First Nation and Mount Currie. Time is one of our most valuable resources. Making a single bead is a huge amount of work. You go hunting, harvest and butcher an animal, clean the bones and the carve them into beads. Or harvest porcupine quills, or specific shells. You’re gathering, preparing and making the bead, and that takes time, energy and resources, even before you start on the actual garment or piece of jewelry. And some beads might require travelling long distances to trade for materials, or are only available seasonally. So, when I bead and gift something to someone, it’s a gift of my time. If I choose to spend my time beading for you it means that you’re worth that huge investment in time, and that I want our relationship to be visible to the wider community. And bringing folks onto the land is the same, it’s an investment and it’s time, a gift that has to be respected. It’s more than two people picking berries, it’s everyone that came before you and me, and everyone that will come after. It’s our ancestors and life experiences and stories. That’s where the acknowledgment ‘all my relations’ comes from, the interconnected relationships with the winged ones, the ones who crawl, the two legged and the four legged, the ones that swim, and the ones who came before. Our connection will have an impact on each other and on the land, for better or for worse.

*Kelsey*: That’s such a beautiful teaching, thank you. And the change for better or for worse is so poignant in the context of relationships between settlers and Indigenous Peoples, it’s complicated, historic, and ongoing.

*Dancing Water*: Exactly. In the urban area close to where I live, there is a lot of trauma and a lot of crime, a lot of people being impacted by the theft of our land and that bag of turnip seed. The impacts of that are still felt. But in that history new relationships can be built. So, including Indigenous Peoples in harvesting isn’t just consultation, isn’t just checking a box the way the government does. Its ongoing relationship and reciprocity. It’s responsibility to one another.

*Kelsey*: Yes! So beautifully said. And I hear what you’re saying about responsibility and reciprocity, but I’m also thinking about the immense strain on resources that our growing population is causing. So, can I honour those relationships by planting Indigenous plants I’m interested in harvesting for myself at home? I mean, I live in a one-bedroom apartment, but let’s pretend I have a big backyard and a garden. If I do that with the intention of reducing the burden on naturally occurring gardens, is that taking on responsibility, or is it shirking it and focusing only on myself?

*Dancing Water*: My Elders have a teaching that you don’t move wild plants. It’s not just about the food source, it’s the relationship with the whole place. So, harvest with good intentions, not in a clear-cut logging method. They aren’t money, they are relationship; don’t sell sage bundles and Indigenous medicines. People are harvesting blossoms from many different plants, because it’s now commonly known that they are medicine, but if you harvest too many, that impacts the bees. They aren’t thinning them; they’re taking all of them. Medicinal teas are a trend—things like mullen, you can use blossoms in a tincture for ear aches, but because that’s being broadcast to a wider audience, non-Indigenous people are taking that and running with it. But it’s only a portion of the information. The recipe will say mullen blossom, but not how to harvest the blossoms or how much to harvest. It all boils down to the intention—is it to mentor? To spread knowledge to people that will respect it? To share or to use it for yourself and make a cup of tea in the winter? But if you’re intention is to make a farm, and to use this plant and the traditional knowledge gifted to you to make money? That’s a troubling relationship. Plants have a spirit, and to move them or rehome them is an interesting concept. They are where they are for a reason, they have a relationship with the land. If you take a clipping and bring it home and it flourishes, this might be a gift for you, but if it dies, that’s a clear answer.

*Kelsey*: That makes perfect sense to me, though to be honest that’s something that I didn’t expect to be talking about in relation to gardening, the growing interest in natural medicines among non-Indigenous peoples, and the marketing of those things for profit. But it brings capitalism into the conversation, which is important. It’s not just resource extraction and climate and wildfires, it’s market-driven pressures.

*Dancing Water*: Yes, there are so many companies that are built around the sale of Indigenous plants, the most obvious example being high end tea stores. People don’t ask where they are getting the plants that make up those teas. What does the chain of custody look like? What intentions went into their harvest? My philosophy and my intentions go into the picking of teas that I make and gift to people, and that can’t be put into a price tag and be sold.

*Kelsey*: What do you mean by chain of custody?

*Dancing Water*: It’s more than custody, it’s really the chain of care. It’s the intentional stages of care that begin with the seed and germination. The acknowledgment for the plant, the harvest, the gifting, the sharing, and for someone to finally ingest it or to put it on their body. It’s the opposite of commodification.

*Kelsey*: This is reminding me of the current focus in the media on the climate crisis and how we, as a global community, need to start thinking about the energy exchanged to get any product from Point A to Point B. I recently had a conversation with a family member about fossil fuels and the hidden carbon footprints that bring vegetables and clothing and car parts through complex global pathways, from initial ingredients to the final consumer, but when you say ‘chain of care’ it’s making me think of energy as intention and human wellbeing. What impact does the food we eat have on the bees, the birds, and other human populations?

*Dancing Water*: Yes, foundationally it’s about the intentions.

*Kelsey:* That reminds me of conversations I’ve had with community members in the territories impacted by the BC wildfires, some of whom have spoken about starting Indigenous gardens closer to home, so that medicine and food plants can be more easily protected during wildfires. Given the intentions of that, to protect plants to sustain relationships with foods and medicines amidst wildfires, does that align with those teachings about leaving plants where they naturally grow?

*Dancing Water*: It’s complicated. Our teachings our dynamic, culture is not fixed, it evolves over time. And also, there’s a whole cultural reasoning for fire, it’s a way to cleanse the territory and the land. So, when Saskatoon bushes were infested with spiders or bugs, or non-Indigenous plants choked out Indigenous plants, fire was used, to thin dense forests, fight disease and target invasive species. And the natural regrowth and the resilience of plants is part of the healing relationship between people, plants, and other animals. It poses the question about the concept of protection: do we need protection from fire or from people as well? This leads us back to complexity of policing plants and food harvesting.

*Kelsey*: So, that’s another layer, that fire has been a tool for ecosystem management and maintaining productive and reciprocal relationships, but has been impeded by colonialism. I’m thinking about pre-colonial Secwépemc land and territory, and how that vast amount of land now includes towns and cities and private property. We’ve restricted the use of fire so much that it has become a dangerous force, as opposed to a productive tool. And logging has resulted in monocrop replanting of Douglas Fir or other trees, taking away the complex diversity of forests that were able to survive and thrive alongside the natural occurrence of fire. And that is one limitation of gardening, in a sense, because it’s a very local action, against a climate crisis caused by systemic and structural forces.

*Dancing Water*: The word in our language for our land is Secwépemcύlecw.

*Kelsey*: Oh, I didn’t know that, thank you!

*Dancing Water*: Of course. And yes, the use of fire has been incredibly impeded by colonialism. Ecosystems are delicately balanced, and that balance has been disrupted. Our hay fields now flood every year, and it’s taking longer and longer for the water to recede. That’s caused by pine beetles, which are spreading because of warmer climates, and are killing trees that then don’t take in water. It’s impacting the natural water table, and then fires kill more trees, which leads to more flooding. Our roads are getting washed out, it’s impacting our ability to move across our territories, and the flooded hay impacts our ability to feed our horses. And it’s changing which plants grow where, because the plants that grow here don’t want to have their roots soaked. Look at what happened in Australia, there’s widespread devastation to the land, to animals and plants, and the people who live there. But you can also see the plants coming back, that resiliency. And maybe that’s a metaphor for the resettling of relationships that we need with the land. To start over, to build up again.

*Kelsey*: Watching the news surrounding the fires in Australia was heart wrenching, and yes, those animals and plants that have not been forced into extinction will come back. I just wonder if we, the global community, will finally learn the lesson. And then where does gardening fit into all this? I do think Euro-Western gardening has its place, and that a small garden in your yard can (re)create a relationship to food and land. But it’s not a one size fits all solution. There’s so much academic literature where individual or community gardening is offered as the way to end food insecurity in Indigenous communities, but that doesn’t take into account the limitations with food security—a Euro-centric measure of health that ignores colonial contexts, and doesn’t capture the intricacies of sovereignty. And that reduced understanding of food as mere nutrition impacts non-Indigenous people as well, we are all supposed to be connected to our food, not buying skinless boneless chicken breasts packed in Styrofoam.

The other side of that is that it can seem easy to say that everyone in Vancouver should have a garden and grow a percentage of their food, it would decrease diet-related disease, increase wellbeing, reduce reliance on global foods. But the ability to garden is also impacted by socio-economic, political and historical forces. Affording a backyard is out of reach for most people, having the time to grow, harvest and prepare fresh produce is a luxury. I think that’s one reason people are turning more to natural food sources, harvesting wild berries, it’s accessible. And people, both Indigenous and non-Indigenous, are craving that connection with nature. I did say that one limitation of gardening is how local it is, but that’s also where the hope can come from. If we can make sure those (re)connections with nature are done in a good way, a lot of good can come out of that.

*Dancing Water*: Yes, there’s a lot of potential, but without addressing income inequality and social injustices, barriers to Euro-Western gardening and to harvesting will continue to be created and upheld. Last year my son and I lived in my travel trailer, and that impacts how you harvest, because you have nowhere to store your food. And more and more people are living in travel trailers, struggling to have a house to live in, let alone the luxury of a garden.

*Kelsey*: It seems like these issues continue to be siloed, housing as separate from food as separate from climate. But if you don’t have a home, or you do but there’s no kitchen, or no outdoor space, or you don’t have rights or access to the land, then your access to land and food sources is diminished, and you’re forced to rely on unsustainable food systems. We really need a holistic view, because it touches on education, housing, healthcare, criminal justice, it’s everything. The project I mentioned earlier, where people in prison grow and gift food, it isn’t just nutrition, it’s potential is to support dignity and belonging, which also requires safe housing, meaningful work, access to social services and recreation, the whole system.

*Dancing Water*: That’s one aspect of Indigenous knowledge systems that exists across diverse Nations, and something my community has been protecting since the beginning of colonialism. When I’m on the land I make an offering, because they surrender to us, they give to us - the plants, the animals. When I’m harvesting those things, because they will be ingested or placed on a person’s body for healing or eating, that chain of care has to be done with a good heart, with good intentions. Because if you do it with a heavy heart or anger, then you will taint that medicine or food, it’s not going to heal or nourish. So those are ways of gardening, it’s holistic and impacts everything. What we can (re)learn is how to all work together, not in isolated harvests, but as a community, an Elder mentoring a youth, or you mentoring someone else. One hand forward one hand back, as Sto:lo Nation scholar and activist Joanne Archibald says. You should always be lifted up by someone and pulling someone up behind you. It’s not individual, it’s communal and relational. We are harvesting for Elders, for people who are struggling with poverty, for people with disabilities. Like I said before, I have four seats in my truck, why would I not fill those seats?

*Kelsey:* I remember a Stol:lo Elder telling me about the importance of cooking with good intentions years ago. We were sitting in a federal prison, and I was interviewing him about prison programs for Indigenous Peoples in Canada. And food wasn’t an explicit part of the conversation, but he saw the connections. And that really changed my life, as dramatic as that sounds. I used to cook professionally, and I love it, it’s my love language. But it can also become a routine, something you do on autopilot after work, when you’re tired or stressed or grumpy. So, bringing that same awareness to the entire food chain. It comes back to that beading teaching you shared, the gift of time. There are so many socio-economic barriers to cooking healthy food at home, but if we fought for the time to cook and eat together, as a right and responsibility, what would that change? It’s so simple, but also really radical.

*Dancing Water*: Yes, exactly! Having awareness of when to harvest, what to harvest and why is important. And the intent and mindset while harvesting should continue, throughout the preparation and use of food and medicines. And to treasure that time. And even if all you can access is vegetables from a grocery store, still treasure the time to cook and to share food. It’s also about complimentary planting and natural occurrences (see: Figure 2). Lemon balm will take over and choke everything out and will overtake natural medicines, so that’s knowledge that’s important to have. Plants such as swamp tea and moss grow well together, both are very useful and can be harvested at the same time. That is no coincidence, and to spend the time to learn that, and to understand those relationships, that can be radical.

*Kelsey*: We keep coming back to relationship, between people, between plants, between colonial structures and systems and things that impede health and wellbeing. So, what do we do with this? Given the climate crises, what does gardening have to offer?

*Dancing Water*: At the moment, Euro-Western gardening projects in Indigenous communities tend to be grant-based, and not sustained by multi-year funding. It has been a challenge to fill the leadership and mentorship roles in the communities I work with. We have community gardens with raised beds, but it’s not clear who has access and who’s in charge. We get seeds donated by different companies, and we can plant them, but sometimes it’s not harmonious, and that leads us back to the chain of care. Someone will take it on for a season, but then they get busy and the plants wither. Euro-Western gardening isn’t a traditional activity for our people, it’s more of a health initiative, and it has potential, but the government health authority and the Band Council keep passing the responsibility back and forth, so no-one wants to take control or steps up when funds are needed for tools, seeds, and ongoing maintenance.

I just learned that my community bought a ranch, we have cows and pigs, and grow potatoes, carrots and other vegetables. Earlier this year my Auntie and I visited and they gave us some vegetables, and later in the year a woman came to my house with pork, because I’m a single mom. They want to have young people go up and help run the garden, they want community involvement. So that’s an example of a community initiative that has a lot of potential. And there should be policy support and funding to ensure it keeps running.

Gardening should be about interconnections, health and awareness. Gardening in a good way is an opportunity to bridge gaps between people, to learn together and possibly blend some Indigenous teachings. I’m part Cree and part Secwépemc, and my son is Tsilhqot’in, so how are my teachings connected to the area that I live in, and where I’m raising my son? There are many layers to Indigenous knowledge systems, and there are many diverse systems that exist across Turtle Island. I’ve learnt some of them. So how do we combine these respectfully? How can we share knowledge to support the health and wellbeing of the land, and all those who live on it? I believe it’s by spending time on the land together, by teaching our children where their food comes from and what that means for their identity. And that can include a ranch, and growing squash and carrots, and harvesting our plant foods and medicines, and hunting and fishing. And this work needs to be supported by policy makers and government funding bodies, through sustainable programming that supports community-led activities on the land, where we teach our children and our youth. Climate scientists should be reaching out, bringing gifts and honoraria payments, and learning from our Elders and Knowledge Keepers, in respectful ways that don’t reproduce colonial harms. And with awareness that Indigenous Peoples are diverse, that different communities have different perspectives, priorities and knowledge systems.

*Kelsey*: Sharing, it’s so simple, but so against what neoliberal capitalism has constructed as the way to live, to be a productive individual, every woman for herself. I know that there’s many food- and land-based projects focused on children and youth across Indigenous communities and urban centres in Canada, some are specific to Indigenous kids, and others are focused on kids from all different backgrounds and ethnic identities, getting them outside, getting their hands in the soil. And I think that’s incredibly important and a real reason to feel some optimism amongst all of this. Sharing knowledge, sharing resources.

*Dancing Water*: Yes, sharing, but with recognition that opportunities to be on the land are being disrupted and restricted by the climate crisis. The fires in my area took a lot away, disrupting animal migration patterns and destroying ecosystems. Not to say it won’t return, or that it didn’t show up in other ways during the evacuations, but there has been a huge impact, and it’s still continuing years later. All our activities on the land had to stop, we were evacuated to hotels in urban areas. Sometimes people had family and friends, sometimes people had nobody. Many of the Nations whose territories we were evacuated onto welcomed us, otherwise we would have been at the mercy of our tiny government evacuation food budget. Some Tsilhqot’in Elders ended up in Abbotsford, and the Sumas First Nation welcomed them and brought them onto the land and shared their foods and medicines. Others, like my Mom, were isolated in other cities and living off hotel food. Those relationships are important in the context of the climate crises, and we need to cultivate them. Right now, I’m looking at our mountain, and it’s still burnt and black, the smaller plants are coming back, but it will take a long time to return to what it was (see: Figure 3). It’s a visual reminder, to care for one another and the land.

*Kelsey*: I remember my first time up in your territory after the fires, and how shocking it was. It’s one thing to see it on the news, but to be there. And then in subsequent years to see that regrowth. I’m also thinking about gender and foodways, your Mom, your Aunt, the women in your life keep coming up. Are there things to be learnt from the diverse Indigenous practices? Or challenges in the way diverse traditions are being taken up? Because patriarchy is tied up with all of this; colonial control of nature, and colonial control of women and gender non-conforming peoples.

*Dancing Water*: There are definitely roles for men and women within food harvesting, for instance menstruation and women, and different First Nations groups have different believes around whether or not that’s appropriate. Some Nations think that the spiritual strength of menstruating women, along with good intentions, can go into the food. Other Nations restrict the contact women have with food during that time, they see it as too powerful and dangerous. But regardless of the belief, there’s a reciprocal role, we are out there using the land together. My son does certain things because he’s taller than me, and has the energy of a young man, not because he was born male. It’s the same reciprocity we have with our natural gardens, we each have our role and responsibility, and each role—that of the hunter, the gatherer, the deer, the plant, are all valued.

*Kelsey:* So, gender roles and relationships should be based in balance and respect. What’s really heartbreaking is to see the evidence of the linkages between violence against nature through aggressive resource extraction and violence against Indigenous women and children.

*Dancing Water*: Yes, the disruption of natural laws is interwoven and complicated. The environment is changing, as are our foodways and relationships to one another. Places that have burnt have pushed animals from different regions. We used to have caribou, then northern logging drove moose down, displacing the caribou. Now with the wildfires, the elk are coming here. There has been considerable impact on the plants and our berries that we count on. The weather is changing, and the growing environment is changing. This is all a result of industry, an increase in harvesters, the change in our climate and regulations around using fire to maintain forest health. There is so much industry in our territories, and it is connected to violence against our women. It’s the breaking of natural laws.

*Kelsey*: There is so much industry up there, I don’t think I’ve ever driven anywhere without getting stuck behind a logging truck, and the impacts of mining have been really extreme. And that makes it very visible. The fires were a couple years ago now, and I think if you live in Vancouver, like I do, or Ottawa, it’s easy to forget, to just move on with your life. The wildfires aren’t here, so we can stop thinking about them. I’m wondering if you’re seeing changes at the community level? Because of how visible the impacts of the fires still are?

*Dancing Water*: In the short term, Red Cross asked people what kind of plants they lost, and they replaced those. My Mom lost red currants, wild raspberries, trampled by firefighting efforts. They also gave out vouchers for food from the butcher in town. My son and I received $200 in total, because we weren’t able to harvest or hunt or fish, and our freezer was empty. But the high-quality butcher shop is expensive, and funds didn’t go far. We made do, but people who were transient couldn’t claim anything. They were told to evacuate, and when they came back, they were unable to access any resources. Today, I don’t know if there is change happening at the community level. I know that Elders are talking about how things were before, when we were able to control burn, when we had more control over our lands. People have evacuation bags ready, something they can grab and leave with. But I think change has to also come from the government. There is funding to support community preparedness, but there hasn’t been widespread community engagement at this point.

*Kelsey*: Are there conversations about supporting vulnerably housed people in the future, if this were to happen again? It seemed, from what I heard from community members, that Elders were really well supported throughout the process, that communities really banded together for them. Can that practice be easily extended?

*Dancing Water*: Things were handled through the natural laws of the land, through our teachings and our ways. It was really beautiful to witness, that Elders were all well supported. My brother, who is in a wheelchair, was taken care of. The community came together and reached out to Elders, single moms, children and people with disabilities and asked ‘what do you need, and how can I help?’ People would check on their family, and then on their sister’s family, and on like that across the kinship system. I think that continuing to support those relationships and those lines of communication is important. We have immense strength in our relationships, and those relationships exist despite ongoing colonialism. To see community members who were previously isolated brought to life by the fires was powerful. To see those folks fighting fires, protecting the homes of those who were evacuated, watching over their pets. It brings us back to how fire was used if plants weren’t doing well. Well some of those folks weren’t doing well and to see them revived and looking after their community is inspiring. I think the wildfires and other extreme weather events like them, as horrible as they are, also catalyze action and create situations in which our teachings can flourish. And policy-makers and leaders need to be supporting this, being present and paying attention when communities comes together, and finding ways to sustain that over time.

*Kelsey*: That’s such a powerful metaphor. And an important aspect of hope in a time where there will be more extreme weather and more natural disasters. I would love to see the same amount of resources that are currently being put into implementing and studying Euro-Western community gardens in Indigenous communities put into environmental protections and learning from the diverse reciprocal and sustainable ways that Indigenous Peoples have been living on this earth for millennia. It was interesting to read the report released by the Tsilhqot’in National Government [27] about their lessons learned during the fires, and plans for the future; they highlight challenges in government-to-government relationships in BC and outlines specific calls to action, which include infrastructure development, governmental and service agreements and land-based stabilization measures, but I’m not sure how much the provincial and federal governments will support that work going forward. As we’ve said earlier, gardens and food are tied up with everything, with governance structures, with housing, with mental health, and the Canadian state isn’t known for engaging well with holism and complexity.

*Dancing Water*: Gabor Maté talks about how the body remembers trauma, and I know that’s true. There is embodied grief that carries forward over time. I’ve felt it in relation to the loss of my father and other things that have happened in my life that align with his passing. And I can let go, it’s difficult but it’s possible, and it involves going out onto the land. It involves giving an offering and not fighting to hold onto sadness. Now, when I spend time in my territory, I see the burnt trees, the ashes and the blackness, you can see the trauma. And it will take many generations for nature to fully heal, but it’s already started. And that’s a visual representation of how long it will take our communities to heal. And there’s progress every day, new plants, new relationships.

*Kelsey*: I think that’s a perfect place to stop, thank you Dancing Water.

## 5. Implications

Diverse Indigenous communities have been raising the alarm about the impacts of colonial resource extraction and climate change for many decades, drawing on reciprocal relationships with varied and complex ecosystems that have been nurtured and sustained since time immemorial. Despite these ongoing and increasingly urgent calls, widespread advocacy and protests, wider Canadian society has been slow to respond. What is often framed as an unfortunate failure in knowledge translation—scientists and media not mobilizing knowledge in ways that catalyze individual, community, and political action [108,109] - the continual erasure of diverse and valid Indigenous sciences, including but not limited to embodied understandings of the natural world and respect for reciprocal balances in nature is a continuation of colonialism through the marginalization of Indigenous epistemologies and ontologies [110,111]. To quote Dian Million, “Indigenous cultures may represent the only living models for different economic and social systems on the planet, ways of life that have the power to challenge capital cultures, even when they are not pure or untouched by capitalism” [112]. Million goes on to describe how different governance structures and ways of living are practiced and imagined in times of emergence and crisis. As Dancing Water articulated above, ways of being together in community were (re)imagined, teachings were upheld and relationships and responsibilities were honoured in ways that colonialism has long fought to upend. In the aftermath of the fires, the importance of relationship to support foodways and land sovereignty highlighted the potentials of gardening, not as small backyard plot, but as responsibility. While there is a great deal of literature on gardening to support Indigenous food security, a large portion is focused on diabetes and other diet-related diseases [57], without attention to the underlying colonial context and responsibilities [20,112,113].

The 2017 BC wildfires provide an opportunity to consider the complex systems that have supported the health and wellbeing of Indigenous Peoples since time immemorial, the structures and forces that are creating barriers to Indigenous and environmental health, and the ways that we can reimagine our relationships to land in the context of climate crises. Indigenous Peoples are not a monolith, and the mechanisms to support health and wellbeing range from engagement with traditional health practices and plant medicines, to Euro-Western nutrition supplements. Just as focus on food security narrows the scope of food-related research, a focus on gardening as the growing of vegetables narrows the potential that engagement with plants, animals, and the wider community has to support health and wellbeing. Gardening in ashes is more than the tilling of soil and the planting of seeds, but the recognition of complex histories and interrelated presents where plant foods and medicines take part in intricate webs of relationship and responsibilities, as well as openness and support for alternate perspectives and relationships to health and food. For many Indigenous communities, (re)connection with ancestral foodways and systems holds potential not only to address food security, but to provide the community cohesion, self-esteem and wellness needed to redress staggering rates of youth mental illness and suicide [23,114] and provide opportunities for community adaptation to climate change [25].

A shift towards food and land as reciprocity and relationship in research and policy agendas and climate negotiations may, in turn, influence correlates of social and health inequities that are continuous with the inequities experienced by Indigenous Peoples [5,115]. Following the leadership of Indigenous communities can provide opportunities to explore community strengths and reinforce Indigenous “land-centered literacies” across generations [116]. Intergenerational knowledge transfer to support food sovereignty, climate activism and community health can create and sustain “transformative alternatives to this [colonial] present” [117] through land-based activities that connect diverse Indigenous Peoples within and across their territories. This dialogue, presented in full, offers a roadmap to exploring relationship and supporting reciprocity within and across research and practice contexts. The implications of listening, asking questions, and collaborating to flatten hierarchies towards supporting climate justice and Indigenous community health are meaningful in relation to historic and ongoing colonialism and cognitive injustices, and the importance of centering Indigenous sciences and Knowledge Keepers in this critical time. Providing a space for Indigenous Peoples to engage in dialogue around foodways and land rights can be an act of decolonization, strengthening communities in the face of systems and policies designed to destroy them [53,114,118]. Food, in all its relationships and intricacies, provides a framework for social learning [4], therefore research and policy on food, climate, and health must include the socio-political, historical and cultural forces at play in food resurgences and ongoing colonial barriers to land, health and wellbeing.

*Kelsey*: Well, that brings us to the end of this paper, and I’m wondering what this means for our work, and what the next steps are for us.

*Dancing Water*: Inevitably, we will continue to have these conversations about the complicated overlaps between food: climate change and colonialism, health and housing and criminal justice, education and the need for awareness.

*Kelsey*: Perhaps somewhere out on the land, on Secwépemcύlecw.

*Dancing Water*: Yes, that sounds perfect. And I also think that we should widen this conversation. This paper is an opportunity to share our thoughts in one way, and to hopefully support conversations with diverse Indigenous Peoples and allies around the world, to widen those transformative dialogues. But to also expand the circle to include more voices from my own community in addition to the surrounding Nations, through a community gathering, sharing food together.

*Kelsey*: I think that’s a great idea. We are two voices, and it would be great to hear from other people with other perspectives. Within the central interior of BC and across the province, there is so much knowledge about foodways and climate change, and I thinking gathering those voices together would be incredibly powerful. With that in mind, I’m wondering if there’s anything else you’d like to add, something to close this conversation?

*Dancing Water*: Yes. When you meet someone, you meet the culmination of their ancestry, their lifetime of experience, everything right up to that moment. Respect that journey, and go forward in a good way. Do not be the source of damage in their life. Now, that teaching can be applied to our land, our Secwépemcύlecw. It’s how you treat other people. It’s how you treat plants and animals. A plant has ancestors that go back far before we were all alive, and it carries knowledge about how to grow to nourish, that needs to be respected and valued (see: Figure 4). Your interactions with plants can leave them better, or far worse. And we can apply that to the whole world around us, to how to go forward.

*Kelsey*: Within Euro-Western and naturally occurring gardens, and in the context of climate change and colonialism.

*Dancing Water*: Yes, go forward in a good way. And recognize the value of our time, and the time and resources of all living things. Recognizing our intentions and shifting our understanding of food and land from one of resource to one of relationship is a very powerful place to stand. When you begin to think about the chain of care that goes into our food, you may realize that there is often not very much care at all, but a severe breakdown with our relationships. But you can find ways to rebuild relationships through reciprocity among neighbours, by trading, by cultivating wild plants and harvesting in responsible ways that honour Indigenous land rights and title, by seeing people as the culmination of their ancestors and experiences, and therefore worthy of respect and compassion. By spending time in gardens. It all can start with a relationship with one single plant.

## Figures and Tables

**Figure 1 ijerph-17-03273-f001:**
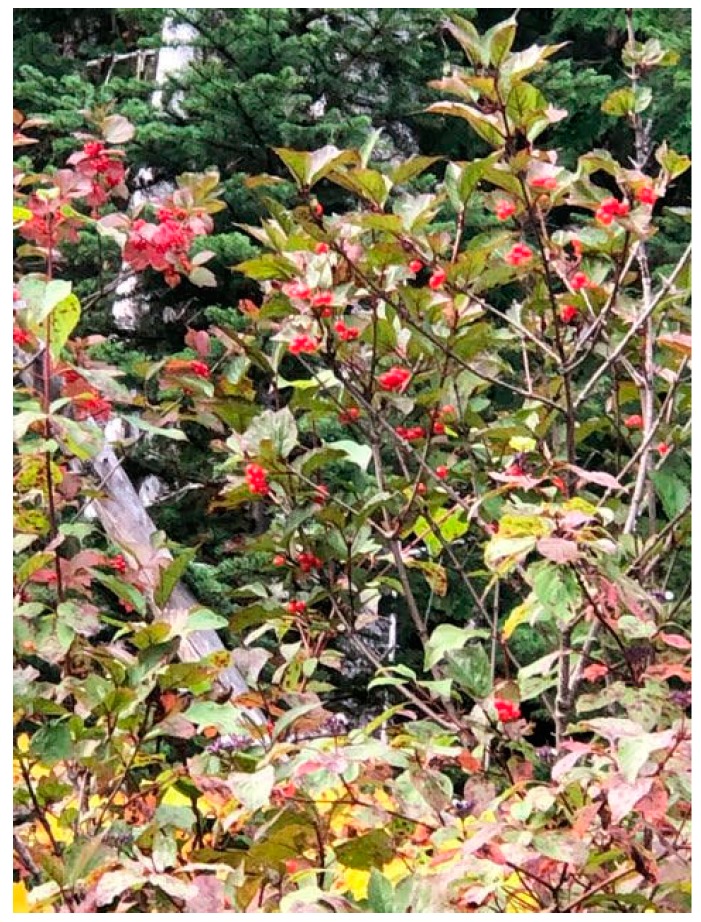
Wild chokecherries.

**Figure 2 ijerph-17-03273-f002:**
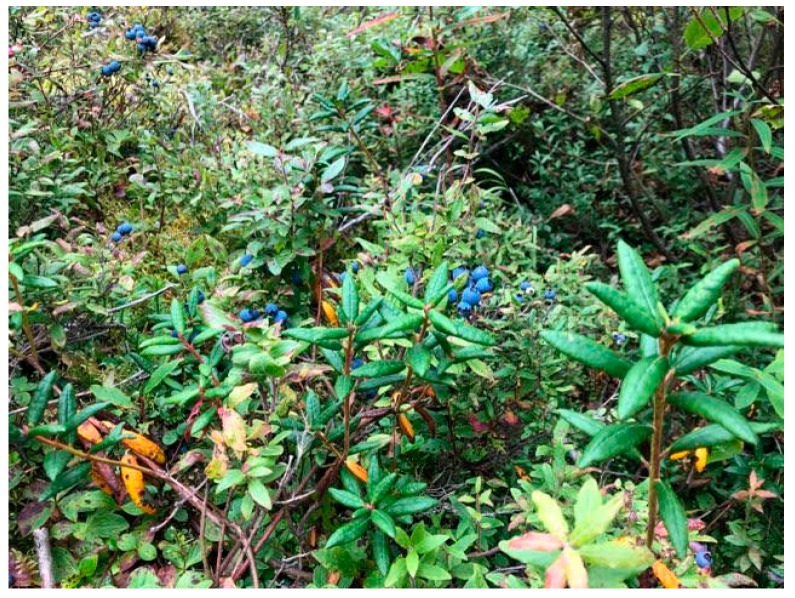
Naturally occurring complimentary crops (low bush blueberry, Labrador tea & moss).

**Figure 3 ijerph-17-03273-f003:**
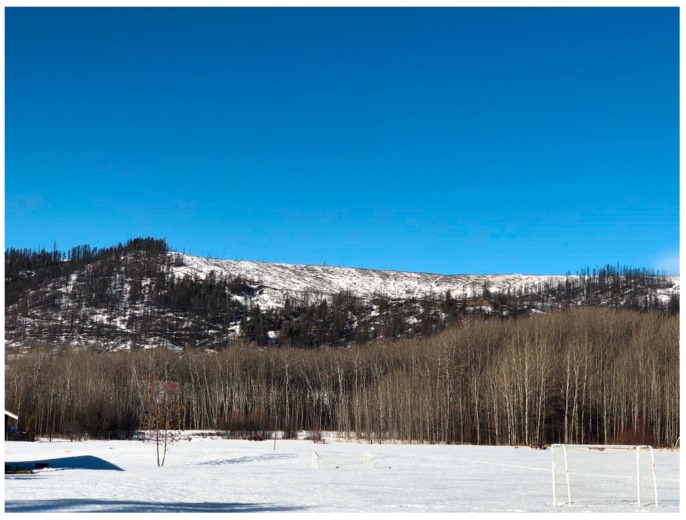
Burnt mountain in Sugarcane reserve, with regrowth.

**Figure 4 ijerph-17-03273-f004:**
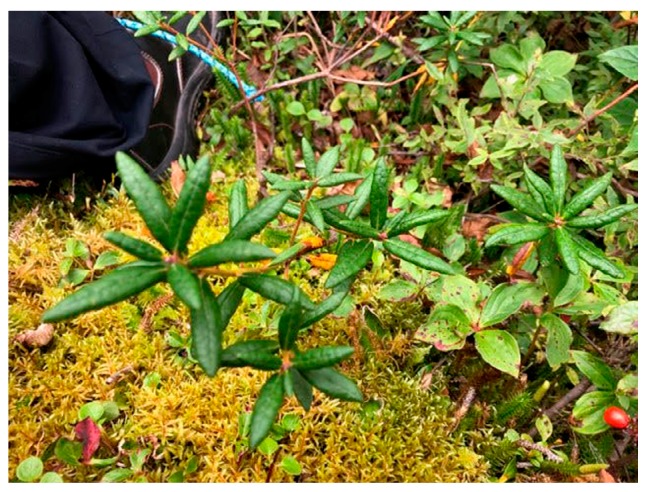
Sitting in relationship, with swamp tea and moss.

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
