# Peer review of "Gardening in Ashes: The Possibilities and Limitations of Gardening to Support Indigenous Health and Well-Being in the Context of Wildfires and Colonialism"

_ijerph, 2020, doi:10.3390/ijerph17093273_

Round 1

Reviewer 1 Report

Overall this paper is well written, with few errors. I appreciate the focus of elevating Indigenous perspectives on gardening, however, I struggle with only highlighting and focusing on one main Indigenous voice throughout the paper. Given that Indigenous communities are collective societies, the paper could have been strengthened with multiple voices and knowledge keepers from the community. Though some of the conversation is interesting, I think that some of the dialogue sections could be trimmed for clarity/focus. Additionally, I did not believe that the abstract accurately set up the paper. Based on the abstract, I was expecting to read a much different paper.

Author Response

February 29, 2020

Dear Editors and Reviewer,

Thank you for your in-depth and thoughtful comments and suggestions regarding our manuscript, entitled Gardening in Ashes: The Possibilities and Limitations of Gardening to Support Indigenous Health and Well-being in the Context of Wildfires and Colonialism. Please see an outline of the changes we have made, based on your feedback, below, and note that all changes are highlighted in yellow in the updated manuscript.

You noted the highlight of one main Indigenous voice throughout the paper, and recommended that, given the collective nature of Indigenous communities, the paper could be strengthened with multiple voices and Knowledge Keepers from the community.

  • In response to this, we added a paragraph (lines 136-144) that clarifies the format of the paper, as a dialogue between two individuals who carry with them the perspectives of their mentors and individuals. This section is highlighted in yellow. We also recognize the limitations of focusing on only two voices and perspectives, given both the collective nature of many Indigenous communities as well as the diversity of perspectives that exist within and across specific communities and Nations. With this in mind, we have added a section at the end of the paper (lines 771-785), that outlines our next steps, which includes continuing our conversations and widening the conversations to include more members of Dancing Water’s community and the surrounding Nations.

You also recommended that the dialogue section could be trimmed for clarity.

  • In response, we have edited down this section, though have not tracked changes. Please let us know if there are specific areas that you think can be edited down further.

Finally, you recommended that the abstract be updated to better reflect the direction that the paper took.

  • In response to this, we have updated the abstract. Changes are aimed at underlining the connections between gardening, tending natural gardens, relationships and reciprocity, and are from lines 8-21.
  •  

Thank you all for your thoughtful feedback, and we look forward to hearing from you in the near future,

Sincerely,

Kelsey Timler & Dancy Water Sandy

Reviewer 2 Report

The title of this paper is slightly misleading.  This is less about garden in ashes then about relationality, community, and food in a colonial landscape.   

Calgary is in Treaty 7 not Treaty 3 territory.  

The discussion about the Blanket Exercise is important.  Consider adding "Blanket Exercise" to the keywords so it can be searched by others.  

There is no ethic and belief that is inherent to any one individual of any race or nation.   Be careful not to fall into the Indigenous vs Non-indigenous duality.  There exists a broad spectrum of assimilation within our communities and a great diversity of thought and beliefs in non-indigenous communities.   The conversation can be better introduced and the implications clarified to address the way this is presented in the dialogue.  

Author Response

March 1, 2020

Dear Editors and Reviewer,

Thank you for your in-depth and thoughtful comments and suggestions regarding our manuscript, entitled Gardening in Ashes: The Possibilities and Limitations of Gardening to Support Indigenous Health and Well-being in the Context of Wildfires and Colonialism. Please see an outline of the changes we have made, based on your feedback, below, and note that all changes are highlighted in yellow in the updated manuscript.

You found the title misleading, noting that “this is less about gardening in ashes then about relationality, community, and food in a colonial landscape.”

  • This distinction between gardening as agricultural practice and relationality, community and food in a colonial landscape is the main thesis of the paper. To clarify this, we have updated the abstract (lines 8-12), as well as added text throughout where the concept of gardening is more explicitly linked to relationships, community and food. Specific text additions are highlighted in yellow and can be found on lines 58, 61-62, 71-73, 85-89, 118-127, 249-250, 350-351, 734-735, 742-754, 755-768, and 792-793.

You noted that Calgary is in Treaty 7 not Treaty 3 territory. 

  • Thank you for catching this typo! We have made this change, which is highlighted in yellow on line 159.

  You saw the discussion on the Blanket Exercise is important and recommended adding "Blanket Exercise" to the keywords so it can be searched by others. 

  • In response to this, we have added blanket exercise to the keywords (line 23).

 Finally, you noted that “there is no ethic and belief that is inherent to any one individual of any race or nation” and to “be careful not to fall into the Indigenous vs Non-indigenous duality. There exists a broad spectrum of assimilation within our communities and a great diversity of thought and beliefs in non-indigenous communities.  The conversation can be better introduced and the implications clarified to address the way this is presented in the dialogue.”  

  • Thank you for this feedback, and for highlighting the ways that our dialogue reduced or simplified the diversity of Indigenous perspectives and thoughts. We have reworked the paper and hope that we have addressed this by adding an explicit paragraph at the beginning of the paper (lines 136-144), which outlines our positions as two individuals with perspectives that are born from our diverse experiences but that do not speak for entire communities. This section, as well as all sections outlined below, have been highlighted in yellow.
  • We have also shifted many of Dancing Water’s statements that were previously written as a collective (e.g., “we go out on the land”), instead phrasing as personal experiences (e.g., “I go out on the land”); specifically, this has been done on lines: 355 (making explicit reference to her community), 380, 381, 536, 537, 597, 615 (making explicit her personal connection to the fires in her region).
  • We have shifted some of Kelsey’s questions/statements towards ‘I’ statements (e.g., from “what do we as settlers need to know” towards “what do I need to know”), this is done on lines 364-365.
  • Additionally, the diversity and complexities of communities has been explicitly layered into the text, specifically on lines 85 (ecosystem-specific knowledge), 120, 380, 381, 597, 604-606, 634, 635, 636-637 (to make explicit diversity as it relates to women and gender non-conforming folks), 701, 718, 720, 724-725, and in the section 742-754.
  • Finally, the final implications of the paper have updated, with added sections highlighted in yellow, outlining the importance of continuing the dialogue and expanding to include more perspectives from diverse Indigenous Peoples (lines 771-783).

Thank you all for your thoughtful feedback, and we look forward to hearing from you in the near future,

Sincerely,

Kelsey Timler & Dancy Water Sandy

Reviewer 3 Report

I just want to start by saying that I think this is an important paper and I look forward to including it as reading in my Indigenous Studies and food studies courses. I think Dancing Water makes a lot of important points about respecting relationships between Indigenous communities and traditional food sources, and this is an important read for health researchers, as well as those developing an interest in foraging and harvesting. That said I have a few suggestions to make parts of this paper more relevant to where I see the most valuable contributions.

The title and abstract, and a good part of the intro focus heavily on fire and gardening. But the paper itself, which it touches significantly on those things, it’s about much more than that. I think the title, abstract, and intro are too limiting relative to what the real contributions of the paper are. The impact of the fires on all of this was significant, but what it really highlighted was the importance of relationships and traditional knowledge and cultural revitalization and not silo-ing knowledge—less about conventional gardens

“widening our understanding of gardening… can support health and healing for indigenous and non indigenous peoples in Canada”

A big part of the take away message for me from this article was not necessarily the importance of gardening conventionally conceived, but the importance of access to wild foods, and the ability to maintain the relationships with those wild foods, as well as relationship-based trading networks. The loving care that goes into maintaining ‘wild’ plants, and how that is disrupted when outsiders without land based teachings come in and harvest everything

Taking care of a plant species in a way that helps a whole ecosystem thrive (as opposed to very controlled Western style gardens)

Your background cites the health benefits of gardening, but I think could use a few citations also about the health benefits of access to wild foods—hunting, foraging grounds, fishing grounds. The health benefits as well as the ways in which communities have been deprived access to these spaces and food communities, and the response of First Nations to fight for this access. Another message that comes out in Dancing Water’s narrative is around the psychological/cultural/physical health impacts of historic trauma as it relates to the land, and the health impacts of disrupted relationships. I think the introduction would benefit from some engagement with that literature as well. Open with what the health sciences community has written about these issues, and then Dancing Water’s words are there to give important qualitative meat to this structure, and provide guidance for where future respectful research should be centered.

The impact of the fires on all of this was significant, but what it really highlighted was the importance of relationships and traditional knowledge and cultural revitalization and not silo-ing knowledge—less about conventional gardens

As far as the format of the article, I think the idea of having a dialogue between a non-Native researcher and an Indigenous culture bearer is really interesting. There were areas where I felt like the chit-chatty nature of the dialogue might turn off more conventional scientifically minded readers (who are among the audience that I think really needs to read this). This article would read very well in an Indigenous Studies course, but I think the information is also so important for readers who are more used to a conventional research article. For example, line 104 “before we jump in, should we introduce ourselves?” or line 115 “always so inspiring, thanks!” or line 617-618 “this is a bit of a tangent but I love tangents!” just as a few examples. I think there are ways to just trim down some of these parts a little bit—to make it clear it’s still a dialogue without the chirpier aspects of the discussion. I, as a female social scientist, appreciate the friendly back and forth and the genuineness in some places. But in other places it feels a little over the top, and again my concern is packaging this really important information and perspective in a way that an out of the ordinary readership for this type of material will consume it.

Author Response

March 1, 2020

Dear Editors and Reviewer,

Thank you for your in-depth and thoughtful comments and suggestions regarding our manuscript, entitled Gardening in Ashes: The Possibilities and Limitations of Gardening to Support Indigenous Health and Well-being in the Context of Wildfires and Colonialism. Please see an outline of the changes we have made, based on your feedback, below, and note that all changes are highlighted in yellow in the updated manuscript.

You noted that “the title and abstract, and a good part of the intro focus heavily on fire and gardening. But the paper itself, which it touches significantly on those things, it’s about much more than that. I think the title, abstract, and intro are too limiting relative to what the real contributions of the paper are. The impact of the fires on all of this was significant, but what it really highlighted was the importance of relationships and traditional knowledge and cultural revitalization and not silo-ing knowledge—less about conventional gardens.”

  • Thank you for your feedback, we have heard from other reviewers as well that the title, abstract and intro are narrowly focused compared with the breadth and depth of the dialogue section. To address this we have text throughout where the concept of gardening is more explicitly linked to relationships, community and food. We have taken your recommendations related to enhancing the literature review in the introduction and added to that section. In addition to this, we have deepened and expanded our literature section to focus on relationship and cultural revitalization. More generally, the distinction between gardening as agricultural practice and relationality, community and food in a colonial landscape have been further outlined in the abstract (lines 8-12), as well as throughout the paper. Specific text additions are highlighted in yellow and can be found on lines 58, 61-62, 71-73, 85-89, 118-127, 249-250, 350-351, 734-735, 742-754, 755-768, and 792-793.

 You also recommended adding citations about the health benefits of access to wild foods – hunting, foraging, fishing, etc., as well as “the ways in which communities have been deprived access to these spaces and food communities, and the response of First Nations to fight for this access. Additionally, you noted that “another message that comes out in Dancing Water’s narrative is around the psychological/cultural/physical health impacts of historic trauma as it relates to the land, and the health impacts of disrupted relationships” and that the introduction would benefit from some engagement with that literature as well.

  • We have added to the introduction and background, providing more context to the wider paper. Specifically, we have added a section describing the ways colonialism has impacted food security (residential schools, industry, agriculture, the reserve system, poverty and related barriers to gas/subsistence tools), on lines 32-37, as well as 85-89.
  • We have added a section detailing the narrow focus of most food security literature and complexity of benefit and relationships for many Indigenous peoples related to foodways, on line 72-73.
  • We have added a paragraph on lines 99-110, which outlines the ways in which contemporary Indigenous Peoples are confronting colonialism and the complex causes of climate change.
  • A paragraph (lines 118- 127) further outlines the differences between Indigenous and Euro-Western concepts of food and health, and Indigenous processes of healing in this context.
  • Lines 755-768 outlines the potentials of “land-centred literacies” in the context of colonialism and climate change, tying together our conversation and the overall themes of resurgence and food sovereignty.
  • While the dietary health impacts of wild foods are widely recognized, the nutritional aspects of gardening and other land-based activities are often focused on within the literature, and we believe that focus on the more relational aspects of foodways is an important contribution to the literature. For this reason, we have respectfully not added citations related to nutritional benefits of wild foods. We believe that that the revisions we have made and detailed above, based on the recommendations related to the cultural, spiritual, etc. impacts of colonialism have strengthen the paper, and add to this focus on our argument for looking beyond nutrition. We hope this makes sense, and would appreciate your thoughts on this, given your previous feedback and our resulting revisions. We look forward to any further input you might have on this particular point.

 Finally, you note that there were areas where “the chit-chatty nature of the dialogue might turn off more conventional scientifically minded readers (who are among the audience that I think really needs to read this). This article would read very well in an Indigenous Studies course, but I think the information is also so important for readers who are more used to a conventional research article.” Specific examples they highlighted were on lines 104, 155 and 617-618.

  • Thank you for this feedback! As a community-based researcher and Knowledge Keeper, we are interested in exploring alternate forms of knowledge mobilization within and outside of academia, and hoped that the informal quality of our dialogue would serve as a form of resistance against academic norms around knowledge sharing and hierarchies. Your thoughtful response, including sharing a bit about your work and your reasoning behind it, has highlighted that we can still work to honor dialogue and relationship in ways that are outside of typical academic conventions, while still providing information in ways that works for researchers across disciplines. We have edited throughout to remove the more “chit chatty” sections of the dialogue, including the specific lines you mentioned above. Again, thank you.

Thank you all for your thoughtful feedback, and we look forward to hearing from you in the near future,

Sincerely,

Kelsey Timler & Dancy Water Sandy